# OpenReview forum: "CharXiv: Charting Gaps in Realistic Chart Understanding in Multimodal LLMs"
_NeurIPS.cc/2024/Datasets_and_Benchmarks_Track — NeurIPS 2024 Track Datasets and Benchmarks Poster_

### Official Review · Reviewer_Bnih · 2024-07-21
**A new benchmark on chart understanding**

**Rating:** 7
**Confidence:** 5

**Review:**

As a new chart understanding benchmark, this work stands out on multiple aspects, including having real world chart images and human annotated questions. The benchmark is also carefully designed, with a rigorous data curation process to ensure high quality data. The authors have conducted extensive experiments and ablations on existing MLLMs which provide valuable analysis and insights. These are the strengths of the work. Potential improvements include providing additional chart metadata, increasing the diversity of chart sources and task types, and adding a non-model-based evaluation protocol.

**Strengths:**

- Comparing to other recent benchmarks relying on synthetic chart images and model-generated questions, this benchmark includes real chart images and human annotated questions, which can better reflect the model's capability to handle real world chart understanding problems.
- The data curation process is well-designed, with an emphasis on the inclusion of diverse chart images, high-quality reasoning questions, and human verified answers.
- The experiments are quite extensive, with results on a large number of most recent MLLMs. The ablations and case studies also provide valuable insights on developing better chart understanding models.

**Additional Feedback:**

N/A

**Clarity:**

The paper is overall well written.
- There seems to be some grammar error / typos in example Q.3

**Correctness:**

- The benchmark can use a better taxonomy. The current distinction between descriptive questions and reasoning questions are not very clear, since the descriptive questions also involve questions that require the model to perform mathematical reasoning such as counting and arithmetic operations.

**Documentation:**

The benchmark is described in sufficient details. The code and data are already available online.
- In addition to the chart categories, it would be great to have chart types as a part of the metadata since they allow further analysis of the model's strengths and weaknesses. Since the benchmark only includes 2K chart images, instead of claiming unbounded chart types, I think it is quite practical and meaningful to perform annotations on chart types.

**Limitations:**

- All chart images are from ArXiv papers. As a result, the topics are mostly STEM-related, making the benchmark less comprehensive.
- The tasks in the benchmark are in the form of VQA with a focus on reasoning. Other important chart understanding tasks, such as chart summarization, chart data extraction and open-ended question answering, are not included.

**Opportunities For Improvement:**

- Currently the descriptive task has limited (19) unique questions. As a result, the questions are not diverse. For example, the benchmark does not include information retrieval questions related to the color of the chart elements (e.g. What is the color of bar whose value is 35?) It would be great to further enrich the diversity of descriptive questions via some human-in-the-loop process similar to the curation of reasoning questions.
- The evaluation protocol can be further improved. Given the short and unambiguous nature of the answers, instead of relying on the GPT-based evaluation which can be cumbersome and costly, the benchmark can benefit from having multiple reference answers (similar to the VQA dataset) to handle the differences in expressing greek symbols and math notations.

**Relation To Prior Work:**

The authors should discuss the following works:
- Kantharaj, Shankar, et al. "OpenCQA: Open-ended Question Answering with Charts." Proceedings of the 2022 Conference on Empirical Methods in Natural Language Processing. 2022.
- Masry, Ahmed, et al. "UniChart: A Universal Vision-language Pretrained Model for Chart Comprehension and Reasoning." Proceedings of the 2023 Conference on Empirical Methods in Natural Language Processing. 2023.

**Summary And Contributions:**

This paper proposes a new chart understanding benchmark ChartXiv, which is based on chart images from ArXiv papers. The benchmark includes templated descriptive questions and human annotated reasoning questions. Experiment results demonstrate the limitations of  existing MLLMs in understanding charts with multiple subplots, answering reasoning questions and identifying unanswerable questions.

---

> ### Author Rebuttal · Authors · 2024-08-16
>
> We thank you for your positive feedback and insightful suggestions! We are encouraged that you find that the dataset to reflect **real-world chart understanding** and **human-annotated questions**, incorporates **high-quality data** through **rigorous curation**. We are also glad to hear that you find the experiments to be **extensive** and the analysis and insights to be **insightful**. We address your comments as follows:
>
> ### Chart-specific descriptive questions.
> We fully agree that enhancing the diversity of descriptive questions would be a valuable improvement. However, questions related to attributes such as color, texture and markers can result in ambiguous answers, so we chose to exclude them.
>
> To ensure unambiguous verification, all descriptive questions are designed to have clear answers, either as written elements present on the chart or as integers. More complex questions involving multiple steps are classified as reasoning questions where we ensure their diversity with human-in-the-loop.
>
> ### Evaluation with multiple references.
> We fully acknowledge your perspective. There are two reasons that prevent us from considering GPT-free auto-validation at the current stage:
>
> 1.  Compared to other VQA tasks, annotating multiple reference answers for CharXiv requires more specialized skills (i.e., writing multiple equivalent expressions of math/greek notations), which creates a much higher barrier for annotators.
>
> 2.  CharXiv is reasoning-heavy, which often leads models to generate chain-of-thought (CoT) responses. Parsing free-form answers from lengthy responses is challenging without relying on an LLM evaluator. This is similar to MathVista, another reasoning-heavy benchmark, where they rely on GPT to extract answers from MLLMs’ responses as a necessary step.
>
>
> ### All charts are STEM-related.
>
> CharXiv is designed to focus purely on chart understanding and reasoning, avoiding the need for advanced domain-specific knowledge. Thus, the key question is whether CharXiv captures a diverse representation of chart images. As shown in Table C of the rebuttal file, CharXiv achieves a **much higher chart diversity score** compared to other chart understanding benchmarks. Regardless, we agree that exploring chart understanding in non-STEM fields could provide more valuable insights.
>
> ### Lack of other chart understanding tasks
> Thank you for the insightful comment! We agree that the tasks you mentioned are essential for chart understanding. Yet, at the current stage of MLLM development, we believe it's more critical to isolate and identify the fundamental atomic abilities that these models may still lack. Tasks like *chart summarization, data extraction and open-ended QA* assess a combination of skills such as *output coherence, structured generation and domain knowledge*. They are undoubtedly valuable, yet they make it difficult to interpret specific gaps in **fundamental chart understanding capabilities**.
>
> CharXiv is designed with questions that focus on information retrieval from the chart and multi-hop reasoning on top of atomic information, allowing for a deep exploration of the reasoning process while *separating it from task-specific skills*.
>
> ### Distinction between descriptive and reasoning questions.
> Thank you for bringing this up! We believe that an intuitive distinction is that reasoning questions usually involve multiple hops and understanding of charts beyond the surface level.
>
> For instance, while a descriptive question might ask the model to count the number of lines, a reasoning question would require the model to count lines *that meet three specific conditions*, involving **multiple steps of logic**. Similarly, a descriptive question might ask for basic arithmetic operations on nearby elements, while a reasoning question would involve performing arithmetic on a variable *across different subplots*, requiring **deeper comprehension**.
>
> These differences highlight that reasoning questions demand more intensive **visual reasoning**, as opposed to the **text-based reasoning** seen in descriptive questions. We are committed to updating our manuscript to reflect this more clearly.
>
> ### Grammar errors and typos
> We sincerely thank the reviewer for checking out our annotated failure cases. We believe that they are crucial in simulating real-world scenarios, where similar mistakes occur frequently in human-LLM interactions. Evaluating model robustness against grammar errors/typos is vital for understanding their practical performance.
>
> ### Related works
> We will update our manuscript with both OpenCQA, a dataset for open-ended chart QA, and UniChart, a model pretrained on multiple chart understanding tasks and finetuned on specific chart datasets.
>
> Moreover, we compared chart image diversity between CharXiv and OpenCQA in Table B of the rebuttal file, and evaluated UniChart in Table A of the rebuttal file. We find that CharXiv encompasses *more diverse charts* compared to OpenCQA, and UniChart finetuned on ChartQA significantly *underperforms* existing MLLMs.
>
> ### Annotation on chart types
>
> We sincerely thank you for this opportunity! We have now included them as part of our dataset.
>
> Our annotation involves querying GPT-4o to acquire all chart types for each chart followed by full manual corrections. Concretely, we pair each chart with a primary chart type as well as other chart types where elements are shared (e.g., line chart as the primary chart with scatter plot and area chart as auxiliary chart types).
>
> We visualized the statistics of chart types in Figure C of the rebuttal file:
> -   distribution of primary chart types (top right)
> -   all observed chart types (top left)
> -   distribution of the number of chart types per chart (bottom left)
> -   a word cloud showing compositions of the “Other” chart type (bottom right).
>
> We are committed to updating the manuscript with analysis on models’ performance on chart types.

---

### Official Review · Reviewer_ZMD4 · 2024-07-23
**Review of Submission172**

**Rating:** 8
**Confidence:** 4
**Correctness:** Correct and sufficient
**Clarity:** The paper is very well-written and ea…

**Review:**

The CharXiv evaluation suite for chart understanding is a significant contribution. The dataset is comprehensive, diverse, and high-quality, the evaluation is extensive, and the findings are insightful.

**Strengths:**

- The CharXiv evaluation suite is large-scale and high-quality: the charts come from papers across eight major subjects and include various types of charts (like pie, bar, line etc). The questions are manually curated by human experts, and the ground-truth answers are validated by hand.

- The paper conducts extensive evaluations of various open-source and proprietary models and reveals a significant disparity between the strongest open-source and proprietary models.

- It provides sufficient evidence to show that existing benchmarks overestimate chart understanding capabilities, highlighting the importance and advantages of CharXiv.

- The analysis of results is thorough, covering the relationship between descriptive and reasoning questions, performance on different numbers of subplots, unanswerable questions, etc. Many analyses are novel, such as those on unanswerable questions and statistics on the number of subplots.

**Additional Feedback:**

NA

**Documentation:**

The collection of datasets is reasonable and sufficient, and the provided dataset and documentation are also well-organized.

**Ethics:**

The paper has no ethical concerns.

**Limitations:**

Limitations of the work is discussed sufficiently in the paper.

**Opportunities For Improvement:**

The automatic grading, while practical and convenient, might introduce some biases and limitations of the GPT-4o model. Cross-checking with human evaluation on some samples might be more convincing.

**Relation To Prior Work:**

The paper provides a clear and concise discussion of related work.

**Summary And Contributions:**

The paper introduces CharXiv, a comprehensive evaluation suite for complex understanding of charts. It consists of 2,323 real-world charts from scientific papers paired with more than 10k questions, including both descriptive and reasoning questions. The paper also conducts an extensive evaluation of different models using this benchmark and reveals several insightful findings, like the clear performance gaps across human, proprietary models and open-source models.

---

> ### Author Rebuttal · Authors · 2024-08-16
>
> We thank you for your positive feedback and constructive comments! We are encouraged that you find the dataset to be **comprehensive, diverse, and high quality**, the evaluation is **extensive**, and the findings are **insightful, thorough, and novel**. Furthermore, we are glad to hear that you find that the **evidence showing that existing benchmarks to overestimate chart understanding capabilities** highlights the importance and advantages of CharXiv. We address your comments as follows:
>
> ### Human evaluations of automatic grading.
> To further validate the feasibility of automatic grading, we performed human grading for a total of 400 questions across descriptive and reasoning tasks on 4 different models. Overall, human grading and GPT-4o based automatic grading match **98.5%** of the time. Moreover, by looking at the matching rate on different models, we do not find any significant favoritism in grading for certain models. We provide detailed statistics in Table C of our rebuttal file.
>
> We hope this resolves your concern and we believe that this further enhances the soundness of our pipeline! We will include relevant statistics in our revised manuscript.

---

> ### Comment · Reviewer_ZMD4 · 2024-08-30
>
> Thanks for the authors' response, the response answers my question. I will keep my positive score for the paper. I have also carefully read the comments from other reviewers and the authors' responses. The answers provided, along with the additional analysis in the attached PDF are helpful. I hope the authors can incorporate these suggestions to further enhance the quality of the paper.

---

> > ### Author Response · Authors · 2024-08-30
> >
> > We thank you again for your feedback and your confirmation of positive assessment! We will ensure that everything we included in the rebuttal will be updated accordingly in our manuscript.

---

### Official Review · Reviewer_vWvR · 2024-07-25
**Review of CharXiv**

**Rating:** 4
**Confidence:** 5
**Correctness:** The dataset construction pipeline see…
**Clarity:** Yes.

**Review:**

Strengths:
1. The paper is clearly written and easy to follow in general.
2. The proposed CharXiv benchmark is useful to verify the model's performance on real-world charts.
3. It points out some of the current model's limitations, such as the poor generalization performance of the open-source models, which can not be reflected with existing benchmarks.
Weaknesses:
1. The chart images of CharXiv are solely from arXiv. Its coverage is quite limited. A comprehensive evaluation benchmark should cover charts from different sources and styles to avoid evaluation bias. Note that existing models have already been trained with charts sourced from arXiv [1].
2. Exising works [2-3] have constructed chart understanding benchmarks based on arXiv papers, which make the contribution of the paper weaker.
3. It lacks evaluation of models that are designed for document understanding [4-5] and chart understanding [6-8]. It would be valuable to examine the generalization ability of the domain-specific models.
4. Though the paper points out that open-source models struggle even with descriptive questions, it lacks an in-depth study of why this can happen, and what can we do to make improvements.

[1] Fanqing Meng, Wenqi Shao, Quanfeng Lu, Peng Gao, Kaipeng Zhang, Yu Qiao, and Ping Luo. 2024. ChartAssisstant: A Universal Chart Multimodal Language Model via Chart-to-Table Pre-training and Multitask Instruction Tuning. arXiv preprint arXiv:2401.02384 (2024). arXiv:2401.02384
[2] Li S, Tajbakhsh N. Scigraphqa: A large-scale synthetic multi-turn question-answering dataset for scientific graphs[J]. arXiv preprint arXiv:2308.03349, 2023.
[3] Hu A, Shi Y, Xu H, et al. mplug-paperowl: Scientific diagram analysis with the multimodal large language model[J]. arXiv preprint arXiv:2311.18248, 2023.
[4] Liu Y, Yang B, Liu Q, et al. Textmonkey: An ocr-free large multimodal model for understanding document[J]. arXiv preprint arXiv:2403.04473, 2024.
[5] Hu A, Xu H, Ye J, et al. mplug-docowl 1.5: Unified structure learning for ocr-free document understanding[J]. arXiv preprint arXiv:2403.12895, 2024.
[6] Han Y, Zhang C, Chen X, et al. Chartllama: A multimodal llm for chart understanding and generation[J]. arXiv preprint arXiv:2311.16483, 2023.
[7] Zhang L, Hu A, Xu H, et al. Tinychart: Efficient chart understanding with visual token merging and program-of-thoughts learning[J]. arXiv preprint arXiv:2404.16635, 2024.
[8] Masry A, Shahmohammadi M, Parvez M R, et al. ChartInstruct: Instruction Tuning for Chart Comprehension and Reasoning[J]. arXiv preprint arXiv:2403.09028, 2024.

**Strengths:**

See Review.

**Additional Feedback:**

None

**Documentation:**

It would be better to provide the exact API version (e.g. gpt-4o-2024-05-13)  used for evaluation for reproducibility.

**Limitations:**

The author includes a limitation section in the paper.

**Opportunities For Improvement:**

1. Include more diverse data sources of the chart image apart from sole arXiv.
2. Add evaluation of document- and chart-specific models.
3. Claim the contribution of the paper more clearly apart from the data contribution.

**Relation To Prior Work:**

Yes.

**Summary And Contributions:**

This paper presents CharXiv, an evaluation benchmark over chart image understanding. It first conducts early experiments and shows that open-source MLLMs are not robust on chart images and questions, which can be attributed to the similarity between the existing test set and training set. For a more comprehensive evaluation, the paper collects 2,323 real-world charts from arXiv papers through a chart  selection pipeline, and annotates the selected charts with both descriptive and reasoning questions by humans. Experiments on CharXiv show that reasoning questions are hard for most MLLMs, and open-source models struggle to answer descriptive questions.
Contributions:
1. Provide a benchmark dataset CharXiv with real-world charts and human-annotated QAs
2. Point out the poor generalization ability of open-source models and limitations of the current evaluations.
3. Provide some insightful analysis over chart understanding, e.g. descriptive abilities are a prerequisite for reasoning.

---

> ### Author Rebuttal · Authors · 2024-08-16
>
> We appreciate your valuable feedback! We are encouraged that you find the benchmark to be **practical** in evaluating chart understanding in **real-world** scenarios and the paper is **clearly written**, points out the **limitations of current open-source models and evaluations**, and the analysis on CharXiv is **insightful**. We address your comments as follows:
>
> ### Charts sources and styles. Models in [1] trained with charts from arXiv.
>
> **arXiv is a single source with diverse charts**
>
> Charts on arXiv are created by the academic community with no standardized styles. We further enhance this diversity by carefully selecting figures with varied color palettes, component placements, layouts, topics, and types. We believe this collection is among the most diverse sets of charts ever curated.
>
> For a visual comparison of CharXiv's diversity against existing benchmarks like ChartQA, please refer to Appendix K. It’s very clear to humans that CharXiv charts are way more diverse.
>
> Quantitatively, we measured the diversity score of embeddings for 500 charts using Inception and DINOv2, showing that CharXiv outperforms all other chart datasets and benchmarks, including ChartQA, which draws from 4 different sources (shown in Table B in the 1-page pdf). We hope this evidence convincingly demonstrates that the number of sources does not equate to the diversity of charts.
>
> **Comparing CharXiv and [1]**
>
> Charts in [1] are **synthesized** from tables in arXiv papers, which tend to lack diversity due to consistent plotting styles. We directly take real chart images from arXiv papers.
>
> We showed in Figure A of the rebuttal file that these charts‘ representations highly resemble PlotQA, another synthetic dataset. In contrast, features of charts from CharXiv form a distinct cluster.
>
> We also evaluated the model trained in [1] in Table A of the rebuttal file, and showed that the model does not necessarily gain an advantage on CharXiv.
>
> ### Comparison to [2, 3]
> We thank the reviewer for bringing these works for comparison.
>
> [2] is a chart understanding dataset & benchmark.
>
> -   For figures, both [2] and CharXiv source charts directly from arXiv papers. CharXiv demonstrates significantly higher chart diversity (Table B in PDF), due to our careful human selection process.
> -   Regarding QAs, [2] focuses on **long-form generation**, requiring models to understand chart contents and *possess expert knowledge*. CharXiv centers on reasoning-intensive questions that don't rely on domain expertise. Consequently, [2] and CharXiv assess different tasks and abilities, making CharXiv a distinctive benchmark of its own.
>
> [3] is a *figure* understanding dataset & benchmark.
>
> -   For figures, both [3] and CharXiv source figures from arXiv papers. CharXiv focuses exclusively on charts, deliberately excluding other figure types like tables, illustrations and natural images, whereas [3] includes a broader range of figures.
> -   Similar to [2], [3] evaluate models on **long-form generation** using captioning metrics, different from CharXiv.
>
> Compared to [2] and [3], we believe CharXiv offers a unique dataset for evaluating models' reasoning abilities **without the need for extensive expert knowledge**. We will incorporate a detailed discussion in our revised manuscript.
>
> ### Evaluations of chart- and document-specific models.
> Thanks for the suggestion! We now included all models [4-8] you mentioned, and present results along with several more baselines in Table A of the pdf. We will also include them in our updated manuscript.
>
> We also present a manual categorization of model errors in Figure A of the rebuttal file.
> -   The strongest models (e.g., DocOwl 1.5, CogAgent) primarily make *perception* errors, where they can localize information but still retrieve incorrect details.
> -   All models struggle with *robustly localizing* the correct information from charts.
> -   The weakest models (e.g., ChartInstruct-FlanT5, TextMonkey) often *hallucinate* or *fail to follow instructions* (i.e., failure).
>
> ### Lack of in-depth study of models’ failure on descriptive questions
> We would like to highlight that we conducted an extensive analysis why open-source models struggle with descriptive questions in our manuscript:
>
> -   L272-L283: We analyze how weaker models often fabricate non-existent elements. We suggested models be trained to discern whether a figure contains the information to answer a question, and to abstain from answering when it does not.
> -   L284-L294: We discovered that open-source models degrade more compared to proprietary ones when handling questions with multiple subplots.
> -   L295-L306: We identify domain-specific performance gaps for descriptive questions.
> -   L920-929: We summarize specific mistakes that are common to both open-source and proprietary models, such as failing to localize, using incorrect elements, making OCR mistakes and more failure modes.
> -   L930-L1169: We provide a detailed case-by-case analysis of these mistakes.
>
> We are happy to provide more information if anything remains unclear.
>
> ### Claim contributions other than data.
> We thank you for pointing out that the contributions apart from data may not have been clearly communicated from our manuscript. As mentioned in your review, our paper reveals the poor generalization of open-source models and limitations of current evaluations and our analysis over model evaluation results on CharXiv provides insightful analysis for the community to better understand the chart understanding as a capability. As suggested by other reviewers, CharXiv also provides novel analysis on unanswerable questions and statistics on the number of subplot and valuable insights on developing better chart understanding models (ZMD4, Bnih). We’ll make sure to claim these contributions more explicitly in our revised manuscript.
>
> ### API version.
> Thanks for the suggestion! We use `gpt-4o-2024-05-13` to perform evaluation and we will include this information in our manuscript.

---

### Author Rebuttal · Authors · 2024-08-16

## General Response
We thank all the reviewers for their detailed feedback and suggestions. We are encouraged by the positive recognition of CharXiv, in particular:

-   **Utility** in evaluating models on **real-world** chart understanding (vWvR, ZMD4, Bnih)
-   **Diversity** in chart images and **high-quality** questions (ZMD4, Bnih)
-   Evidence on the **limited generalization** of chart understanding in open-source models and **over-estimation of performance** from existing evaluations (vWvR, ZMD4)
-   **Extensive evaluations** on different models (ZMD4, Bnih)
-   **Insightful and novel analysis** of model performance (vWvR, ZMD4, Bnih)

We have now added the following additional results and analysis:
-   Evaluation results on **12** domain-specific models.
-   Error categorization on **650** model response pairs.
-   Data diversity evaluation on **9** datasets.
-   Human evaluation of automatic grading on **400** samples.
-   Chart type categorization on all **2,323** charts.

In the provided PDF, we have included the following content based on the reviewers’ suggestions:
-   Table A: Evaluation results of 12 chart- and document-specific models (vWvR)
-   Figure A: Error categorization of chart- and document-specific models (vWvR)
-   Figure B: Comparisons between CharXiv and ChartAssistant (vWvR)
-   Table B: Comparisons between CharXiv and other chart- and figure-related datasets (vWvR)
-   Table C: Human evaluation of automatic evaluation (ZMD4)
-   Figure C: Categorizations of chart types (Bnih)

We have taken all of the reviewers’ comments into account and we believe that these improvements **significantly  enhanced the quality of our manuscript and the soundness of our work.**

---

### Author Response · Authors · 2024-08-21
**Gentle Reminder: Request for Feedback on Rebuttal**

Dear Reviewers,

We are writing to kindly follow up on the rebuttal we submitted last Friday. We understand that everyone has busy schedules, and we sincerely appreciate the time and effort you have already dedicated to reviewing our work.

If possible, we would greatly appreciate any feedback, comments, or actions you might have regarding the rebuttal. Your insights are incredibly valuable to us, and we are eager to address any remaining questions or concerns if there exists any.

Thank you very much for your time and continued consideration!

Best,

Paper #172 Authors

---

### Author Response · Authors · 2024-08-30

Dear Reviewers,

We hope this message finds you well. As tomorrow is the last day of the discussion period, we kindly request your feedback on our rebuttal. We have not received any feedback or responses over the past two weeks, which has been a bit discouraging.

To reiterate, we have carefully addressed all the concerns raised in your initial reviews. We would greatly appreciate any additional feedback or confirmation that your concerns have been resolved.

Best regards,

Paper #172 Authors

---

### Decision · Program_Chairs · 2024-09-26

**Decision:**

Accept (Poster)

**Comment:**

The paper presented CharXiv, a comprehensive dataset for evaluating understanding of charts.   The dataset is quite rich and the authors have included detailed evaluations of many LLMs on this dataset and shared their findings on the performance gaps between human and LLMs.  The authors also carefully answered reviewers' questions.   Please revise the manuscripts accordingly as the readers might have the similar questions.